# Positive Periodic Solution for Second-Order Nonlinear Differential Equations with Variable Coefficients and Mixed Delays

**DOI:** 10.3390/e24091286

**Published:** 2022-09-12

**Authors:** Zejian Dai, Bo Du

**Affiliations:** 1School of Mathematics and Statistics, Chaohu University, Chaohu 238024, China; 2School of Mathematics and Statistics, Huaiyin Normal University, Huaian 223300, China

**Keywords:** fixed point theorem, positive periodic solution, existence, neutral, 34C25, 34K40

## Abstract

In this paper, we study two types of second-order nonlinear differential equations with variable coefficients and mixed delays. Based on Krasnoselskii’s fixed point theorem, the existence results of positive periodic solution are established. It should be pointed out that the equations we studied are more general. Therefore, the results of this paper have better applicability.

## 1. Introduction

The main purpose of this paper is to consider positive periodic solution for two classes of second-order nonlinear differential equations with variable coefficients and mixed delays as follows:(1)$$x^{\prime \prime }\left(t\right)+b\left(t\right)x^{\prime }\left(t\right)+a\left(t\right)x\left(t\right)=f\left(x\left(t-\delta \left(t\right)\right)\right)+\int_{0}^{\infty }k\left(s\right)h\left(x\left(t-s\right)\right)ds$$
and
(2)$${\left(Ax\left(t\right)\right)}^{\prime \prime }+b\left(t\right)x^{\prime }\left(t\right)+a\left(t\right)x\left(t\right)=f\left(x\left(t-\delta \left(t\right)\right)\right)+\int_{0}^{\infty }k\left(s\right)h\left(x\left(t-s\right)\right)ds,$$
where a,b,δ∈C(R,(0,∞)) are T− periodic functions, f,h∈C(R,R),
(3)(Ax)(t)=x(t)−c(t)x(t−τ),
$c\left(t\right)\in C^{1}\left(R,R\right)$ is an T− periodic function with |c(t)|≠1, τ>0 is a constant, and k(s) is a continuous and integrable function on [0,∞) with $\int_{0}^{\infty }k\left(s\right)ds=1$.

Equation (Equation 1) is a non-neutral second-order nonlinear differential equation which has received much attention. Wang, Lian, and Ge [1] studied the following second-order differential equation with periodic boundary conditions:(4)$$\left\{\begin{array}{c} x^{\prime \prime }\left(t\right)+p\left(t\right)x^{\prime }\left(t\right)+q\left(t\right)x\left(t\right)=h\left(t\right)\\ x\left(0\right)=x\left(\omega \right),\;\;x^{\prime }\left(0\right)=x^{\prime }\left(\omega \right). \end{array}\right.$$
In Equation (Equation 4), the periodic solution $x\left(t\right)=\int_{t}^{t+\omega }G\left(t,s\right)h\left(s\right)ds$ exists, where
(5)$$G\left(t,s\right)=\frac{\int_{t}^{s}\exp\left[\int_{t}^{u}b\left(v\right)dv+\int_{u}^{s}a\left(v\right)dv\right]du+\int_{s}^{t+\omega }\exp\left[\int_{t}^{u}b\left(v\right)dv+\int_{u}^{s+\omega }a\left(v\right)dv\right]du}{\left[\exp\left(\int_{0}^{\omega }a\left(u\right)du\right)-1\right]\left[\exp\left(\int_{0}^{\omega }b\left(u\right)du\right)-1\right]}.$$
They obtained G(t,s)>0 for t,s∈[0,ω] if the following conditions are satisfied:

(A_1_)There are continuous ω-periodic functions a(t) and b(t) such that $\int_{0}^{\omega }a\left(t\right)dt>0,$$\int_{0}^{\omega }b\left(t\right)dt>0$ and
$$a\left(t\right)+b\left(t\right)=p\left(t\right),\;\;b^{\prime }\left(t\right)+a\left(t\right)b\left(t\right)=q\left(t\right)\;\mathrm{for}\;t\in R;$$
(A_2_)

$${\left(\int_{0}^{\omega }p\left(u\right)du\right)}^{2}\geq 4\omega^{2}\exp\left(\frac{1}{\omega }\int_{0}^{\omega }\ln q\left(u\right)du\right).$$



Obviously, G(t,s) in (5) is too complex, and the conditions for satisfying G(t,s)>0 are too strong and cannot be easily used. Bonheure and Torres [2] studied the existence of positive solutions for the model scalar second-order boundary value problem
(6)$$\left\{\begin{array}{c} -u^{\prime \prime }+c\left(x\right)u^{\prime }+a\left(x\right)u=\frac{b(x)}{u^{p}\left(x\right)},\;x\in R,\\ \lim_{|x|\to \infty }u\left(x\right)=0, \end{array}\right.$$
where a,b,c>0 are locally bounded coefficients and p>0. For (6), the authors also obtained the Green function, which can be used for studying the homoclinic solution and bounded solution of a second-order singular differential equation. However, it is inconvenient to use this Green function to study the periodic solutions of (1). In order to overcome the above difficulties, we use the order reduction method for studying periodic solutions of (1) in the present paper. For more results about second-order singular differential equation with variable coefficients and delays, see, e.g., [3,4,5,6,7] and cited references.

Equation (Equation 2) is a neutral second-order nonlinear differential equations. Periodic solutions of higher-order differential equations have a wide range of applications, and many researchers have conducted a lot of research on them. Liu and Huang [8] studied the existence and uniqueness of periodic solutions for a kind of second-order neutral functional differential equations. Lu and Ge [9] considered periodic solution problems for a kind of second-order differential equation with multiple deviating arguments. Luo, Wei, and Shen [10] investigated the existence of positive periodic solutions for two kinds of neutral functional differential equations. Arbi, Guo, and Cao [11] studied a novel model of high-order BAM neural networks with mixed delays in the Stepanov-like weighted pseudo almost automorphic space. Xin and Cheng [12] studied a third-order neutral differential equation. In [13], the authors considered the existence of periodic solutions for a p-Laplacian neutral functional differential equation by using Mawhin’s continuation theorem. For more recent results about positive periodic solutions of neutral nonlinear differential equations, see, e.g., [14,15,16,17,18]. We found that the results of existing positive periodic solutions mostly depend on Green functions and the properties of neutral operator. However, it is very difficult to obtain proper Green functions. In this paper, we develop some new mathematical methods for obtaining the existence of positive periodic solutions without using Green functions. It should be pointed out that, in 2009, we obtained an important result (see the below Lemma 1) for the properties of neutral operator which can be easily used to study the periodic solution problems of functional differential equations. This paper is devoted to studying the existence for positive periodic solutions of Equations (Equation 1) and (Equation 2) by using the Krasnoselskiis fixed point theorem and some mathematical analysis techniques. The main contributions of this paper are listed as follows:(1)Equations (1) and (2) in the present paper are more general, including the existing classical second-order differential equations, than the considered equations in [1,4,5,6,7,15,16,17,18]. Therefore, the results of this paper are more general and better applicable.(2)Since it is very difficult to obtain Green functions of second-order nonlinear differential equations with variable coefficients, we develop new methods for overcoming the above difficulties. Using appropriate variable transformation, we transform a second-order equation into an equivalent one-dimensional system, so we do not need to solve the Green function. The research method of this paper is different from the existing research methods, see, e.g., [1,15,16,17,18].(3)In 2009, we obtained the important properties of the neutral operator in [19]. In the past, we mostly used this important property to study the existence of periodic solutions. In this paper, we used this important property to study the existence of positive periodic solutions for the first time.

The following sections are organized as follows: Section 2 gives the main lemmas. Section 3 gives the existence results of positive periodic solutions to Equation (Equation 1). Section 4 gives the existence results of positive periodic solutions to Equation (Equation 2). In Section 5, an example is given to show the feasibility of our results. Finally, Section 6 concludes the paper.

## 2. Main Lemmas

Denote $f_{0}=\max_{t\in R}\left|f\left(t\right)\right|,\;C_{T}=\left\{x:x\in C\left(R,R\right),x\left(t+T\right)\equiv x\left(t\right)\right\},\;C_{T}^{1}=\left\{x:x\in C^{1}\left(R,R\right),x^{\prime }\in C_{T}\right\}$; *T* is a given positive constant.

**Lemma** **1**([19]). *Let*
$$A:C_{T}\to C_{T},\;\;\left[Ax\right]\left(t\right)=x\left(t\right)-c\left(t\right)x\left(t-\tau \right),\;\;\forall t\in R.$$
*If |c(t)|≠1, then operator A has continuous inverse $A^{-1}$ on $C_{T}$, satisfying*
*(1)* $$\left[A^{-1}f\right]\left(t\right)=\left\{\begin{array}{cc} f\left(t\right)+\sum_{j=1}^{\infty }\prod_{i=1}^{j}c\left(t-\left(i-1\right)\tau \right)f\left(t-j\tau \right),\;c_{0}<1,\;\forall f\in C_{T}, & \\ -\frac{f(t+\tau )}{c(t+\tau )}-\sum_{j=1}^{\infty }\prod_{i=1}^{j+1}\frac{1}{c(t+i\tau )}f\left(t+j\tau +\tau \right),\;\sigma >1,\;\forall f\in C_{T}, & \end{array}\right.$$*(2)* $$\int_{0}^{T}\left|\left[A^{-1}f\right]\left(t\right)\right|dt\leq \left\{\begin{array}{cc} \frac{1}{1-c_{0}}\int_{0}^{T}\left|f\left(t\right)\right|dt,\;c_{0}<1,\;\forall f\in C_{T}, & \\ \frac{1}{\sigma -1}\int_{0}^{T}\left|f\left(t\right)\right|dt,\;\sigma >1,\;\forall f\in C_{T}, & \end{array}\right.$$*(3)* $$|A^{-1}{f|}_{0}\leq \left\{\begin{array}{cc} \frac{1}{1-c_{0}}{\left|f\right|}_{0},\;c_{0}<1,\;\forall f\in C_{T}, & \\ \frac{1}{\sigma -1}{\left|f\right|}_{0},\;\sigma >1,\;\forall f\in C_{T}, & \end{array}\right.$$*where $c_{0}=\max_{t\in [0,T]}|c\left(t\right)|,\;\sigma =\min_{t\in [0,T]}\left|c\left(t\right)\right|.$*

**Definition** **1**([20]). *Let X be a Banach space and K be a closed, nonempty subset of X. K is a cone if*
*(i)* *αu+βv∈K for all u,v∈K and α,β≥0,**(ii)* *u,−u∈K imply u=0.*

**Lemma** **2**([21] (Krasnoselskii’s fixed point theorem)). *Let B be a Banach space and K be a cone in B. Assume that $\Omega_{1}$ and $\Omega_{2}$ are open bounded subsets of B with $0\in \Omega_{1}\subset {\bar{\Omega }}_{1}\subset \Omega_{2},$ and let*
$$T:K\cap ({\bar{\Omega }}_{2}\setminus \Omega_{1})\to K$$
*be a completely continuous operator such that either*
$$\left||Tu|\right|\leq ||u||,\;\;u\in K\cap \partial \Omega_{1}\;\mathrm{and}\;||Tu||\geq ||u||,\;\;u\in K\cap \partial \Omega_{2},$$
*or*
$$\left||Tu|\right|\geq ||u||,\;\;u\in K\cap \partial \Omega_{1}\;\mathrm{and}\;||Tu||\leq ||u||,\;\;u\in K\cap \partial \Omega_{2},$$
*Then, T has a fixed point in $K\cap ({\bar{\Omega }}_{2}\setminus \Omega_{1})$.*

## 3. Positive Periodic Solution of Equation (Equation 1)

Let
$$y\left(t\right)=x^{\prime }\left(t\right)+\xi x\left(t\right),$$
where ξ>0 is a constant. Then, Equation (Equation 1) is changed into the following system:(7)$$\left\{\begin{array}{c} x^{\prime }\left(t\right)=-\xi x\left(t\right)+y\left(t\right),\\ y^{\prime }\left(t\right)=-\left(b\left(t\right)-\xi \right)y\left(t\right)-\left[a\left(t\right)-\left(b\left(t\right)-\xi \right)\xi \right]x\left(t\right)+f\left(x\left(t-\delta \left(t\right)\right)\right)+\int_{0}^{\infty }k\left(s\right)h\left(x\left(t-s\right)\right)ds. \end{array}\right.$$
Since system (7) is equivalent to (1), we just have to study the existence of positive periodic solutions to system (7).

Let
$$X=\{z\in C\left(R,R^{2}\right):z\left(t+T\right)=z\left(t\right),\;z={\left(x,y\right)}^{T}\}$$
with the norm $\left||z||=\max\{|x{|}_{0},|y{|}_{0}\right\}$, where ${\left|f\right|}_{0}=\max_{t\in R}\left|f\left(t\right)\right|$. Then, *X* is a Banach space. Throughout this paper, we need the following assumption:

(H${}_{1}$) b(t)−ξ>0 for t∈R,

where ξ>0 is defined by (7). Let
$$K=\{z={\left(x,y\right)}^{T}\in X:x\left(t\right)\geq \theta |x{|}_{0}\;\mathrm{and}\;y\left(t\right)\geq \theta |y{|}_{0},\;t\in \left[0,T\right]\}$$
where $\theta =\min\{\frac{\overset{ˇ}{g}}{\hat{g}},\frac{\overset{ˇ}{\hbar }}{\hat{\hbar }}\}$, $\overset{ˇ}{g},\hat{g},\overset{ˇ}{\hbar },\hat{\hbar }$ are defined by (Equation 10) and (Equation 11). Integrate (7) from *t* to t+T and obtain that
(8)$$\left\{\begin{array}{c} x\left(t\right)=\int_{t}^{t+T}g\left(t,s\right)y\left(s\right)ds,\\ y\left(t\right)=\int_{t}^{t+T}\hbar \left(t,s\right)F\left(s\right)ds, \end{array}\right.$$
where
$$g\left(t,s\right)=\frac{e^{(s-t)\xi }}{e^{\xi T}-1},$$
$$\hbar \left(t,s\right)=\frac{\exp\left(\int_{t}^{s}\left(b\left(u\right)-\xi \right)du\right)}{\exp\left(\int_{0}^{T}\left(b\left(u\right)-\xi \right)du\right)-1},$$
$$F\left(s\right)=-\left[a(s)-(b(s)-\xi )\xi \right]x\left(s\right)+f\left(x\left(s-\delta \left(s\right)\right)\right)+\int_{0}^{\infty }k\left(u\right)h\left(x\left(s-u\right)\right)du.$$
It is easy to see that
(9)g(t+T,s+T)=g(t,s),ℏ(t+T,s+T)=ℏ(t,s).
By assumption (H${}_{1}$), we have
(10)$$\overset{ˇ}{g}=:\frac{e^{-T\xi }}{e^{\xi T}-1}\leq g\left(t,s\right)\leq \frac{e^{T\xi }}{e^{\xi T}-1}=:\hat{g}$$
and
(11)$$\overset{ˇ}{\hbar }=:\frac{\exp\left({-T|b\left(t\right)-\xi |}_{0}\right)}{\exp\left(\int_{0}^{T}\left(b\left(u\right)-\xi \right)du\right)-1}\leq \hbar \left(t,s\right)\leq \frac{\exp\left({T|b\left(t\right)-\xi |}_{0}\right)}{\exp\left(\int_{0}^{T}\left(b\left(u\right)-\xi \right)du\right)-1}=:\hat{\hbar }.$$
For each $z={\left(x,y\right)}^{T}\in X$, define an operator Φ:K→K as
$$\begin{array}{cc} \left(\Phi z)(t\right) & ={\left(\int_{t}^{t+T}g\left(t,s\right)y\left(s\right)ds,\int_{t}^{t+T}\hbar \left(t,s\right)F\left(s\right)ds\right)}^{T}, \end{array}$$
where $\Phi z={\left(\Phi_{1}z,\Phi_{2}z\right)}^{T},\;g\left(t,s\right),\hbar \left(t,s\right)$, and F(s) are defined by (Equation 8). Thus, the existence of a positive periodic solution of system (Equation 7) is equivalent to finding the fixed point of operator Φ.

**Lemma** **3.**
*The mapping Φ maps K into K.*


**Proof.** For each z∈K, $\left(\Phi_{1}z\right)\left(t\right)$ and $\left(\Phi_{2}z\right)\left(t\right)$ are continuous in t∈[0,T]. By (Equation 9) we obtain
$$\begin{array}{cc} \left(\Phi_{1}z\right)\left(t+T\right) & =\int_{t+T}^{t+2T}g\left(t+T,s\right)y\left(s\right)ds\\ & =\int_{t}^{t+T}g\left(t+T,s+T\right)y\left(s\right)ds\\ & =\int_{t}^{t+T}g\left(t,s\right)y\left(s\right)ds=\left(\Phi_{1}z\right)\left(t\right) \end{array}$$
and
$$\begin{array}{cc} \left(\Phi_{2}z\right)\left(t+T\right) & =\int_{t+T}^{t+2T}\hbar \left(t+T,s\right)F\left(s\right)ds\\ & =\int_{t}^{t+T}\hbar \left(t+T,s+T\right)F\left(s\right)ds\\ & =\int_{t}^{t+T}\hbar \left(t,s\right)F\left(s\right)ds=\left(\Phi_{2}z\right)\left(t\right). \end{array}$$
Hence, (Φz)(t+T)=(Φz)(t) and Φz∈X. By (3.4), for z∈K we have
$$|\Phi_{1}{z|}_{0}\leq \hat{g}\int_{0}^{T}\left|y\left(s\right)\right|ds$$
and
(12)$$\left(\Phi_{1}z\right)\left(t\right)\geq \overset{ˇ}{g}\int_{0}^{T}|y\left(s\right)|ds\geq \frac{\overset{ˇ}{g}}{\hat{g}}{\left|\Phi_{1}z\right|}_{0}.$$
Similarly, by (Equation 11), for z∈K we have
(13)$$\left(\Phi_{2}z\right)\left(t\right)\geq \overset{ˇ}{\hbar }\int_{0}^{T}|F\left(s\right)|ds\geq \frac{\overset{ˇ}{\hbar }}{\hat{\hbar }}{\left|\Phi_{2}z\right|}_{0}.$$
In view of (Equation 12) and (Equation 13), we have $\left(\Phi_{1}z\right)\left(t\right)\geq \theta {\left|\Phi_{1}z\right|}_{0}$ and $\left(\Phi_{2}z\right)\left(t\right)\geq \theta {\left|\Phi_{2}z\right|}_{0}$. Hence, Φz∈K.□

**Lemma** **4.**
*The mapping Φ:K→K is completely continuous.*


**Proof.** We first show the operator Φ is continuous. Since f(x) is continuous, it is easy to see that, for any L,ε>0, there exists δ>0 such that for each γ,ρ∈X, where $\gamma ={\left(\gamma_{x},\gamma_{y}\right)}^{T},\;\rho ={\left(\rho_{x},\rho_{y}\right)}^{T}$, ||γ||≤L,||ρ||≤L, and ||γ−ρ||<δ imply
(14)$$\underset{u\in R}{\sup}\left|\gamma_{y}\left(u\right)-\rho_{y}\left(u\right)\right|\leq \frac{\varepsilon }{2T\hat{g}}$$
and
(15)$$\begin{array}{cc} & \underset{s\in R}{\sup}{\left(|a\left(s\right)-\left(b\left(s\right)-\xi \right)\xi \right|}_{0}\left|\gamma_{x}-\rho_{x}\right|\\ \; & +|f\left(\gamma_{x}\left(s-\delta \left(s\right)\right)\right)-f\left(\rho_{x}\left(s-\delta \left(s\right)\right)\right)|+\int_{0}^{\infty }\left|h\left(\gamma_{x}\left(s-u\right)\right)-f\left(\rho_{x}\left(s-u\right)\right)\right|du)\\ \; & \leq \frac{\varepsilon }{2T\hat{\hbar }}. \end{array}$$
From (Equation 14) and (Equation 15), if p,q∈K with $p={\left(p_{x},p_{y}\right)}^{T},\;q={\left(q_{x},q_{y}\right)}^{T},\;||p||\leq L,||q||\leq L$, and ||p−q||<δ, then
$$\begin{array}{c} |\Phi_{1}p-\Phi_{1}{q|}_{0}\leq \hat{g}\int_{0}^{T}\left|p_{y}\left(s\right)-q_{y}\left(s\right)\right|ds<\frac{\varepsilon }{2} \end{array}$$
and
$$\begin{array}{cc} |\Phi_{2}p-\Phi_{2}{q|}_{0} & \leq \hat{\hbar }\int_{0}^{T}{\left(|a\left(s\right)-\left(b\left(s\right)-\xi \right)\xi \right|}_{0}\left|p_{x}-q_{x}\right|\\ & +|f\left(p_{x}\left(s-\delta \left(s\right)\right)\right)-f\left(q_{x}\left(s-\delta \left(s\right)\right)\right)|+\int_{0}^{\infty }k\left(u\right)\left|h\left(p_{x}\left(s-u\right)\right)-h\left(q_{x}\left(s-u\right)\right)\right|du)ds\\ & <\frac{\varepsilon }{2}. \end{array}$$
This yields
$$||\Phi p-\Phi q||\leq \max\{|\Phi_{1}p-\Phi_{1}q{|}_{0}+|\Phi_{2}p-\Phi_{2}q{|}_{0}\}<\varepsilon .$$
Hence, Φ is continuous.Next, we show that Φ is uniformly bounded. Let ε=1. Using the above proof, we know that for any μ>0,p,q∈K with ||p||≤μ,||q||≤μ, there exists η>0 such that ||p−q||≤η imply
(16)$$|p_{y}\left(s\right)-q_{y}\left(s\right)|<1$$
and
(17)$$\begin{array}{cc} & {\left|a\left(s\right)-\left(b\left(s\right)-\xi \right)\xi \right|}_{0}|p_{x}-q_{x}|+|f\left(p_{x}\left(s-\delta \left(s\right)\right)\right)-f\left(q_{x}\left(s-\delta \left(s\right)\right)\right)|\\ \; & +\int_{0}^{\infty }k\left(u\right)\left|h\left(p_{x}\left(s-u\right)\right)-h\left(q_{x}\left(s-u\right)\right)\right|du<1. \end{array}$$
Choose a positive number *N* such that $\frac{\mu }{N}<\eta .$ Let $z={\left(z_{1},z_{2}\right)}^{T}\in K$ and $z^{i}\left(t\right)=\frac{z(t)i}{N}$ for i=0,1,⋯,N. If ||z||≤μ, then
(18)$$\begin{array}{cc} \left|\right|z^{i}-z^{i-1}\left|\right| & =\max\left\{\underset{t\in R}{\sup}|\frac{z_{1}\left(t\right)i}{N}-\frac{z_{1}\left(t\right)\left(i-1\right)}{N}|,\;\underset{t\in R}{\sup}\left|\frac{z_{2}\left(t\right)i}{N}-\frac{z_{2}\left(t\right)\left(i-1\right)}{N}\right|\right\}\\ \; & =\frac{1}{N}\left||z|\right|\leq \frac{\mu }{N}<\eta . \end{array}$$
From (Equation 16)–(Equation 18), we have
(19)$$|z_{2}^{i}\left(s\right)-z_{2}^{i-1}\left(s\right)|<1$$
and
(20)$$\begin{array}{cc} & {\left|a\left(s\right)-\left(b\left(s\right)-\xi \right)\xi \right|}_{0}|z_{1}^{i}\left(s\right)-z_{1}^{i-1}\left(s\right)|+|f\left(z_{1}^{i}\left(s-\delta \left(s\right)\right)\right)-f\left(z_{1}^{i-1}\left(s-\delta \left(s\right)\right)\right)|\\ \; & +\int_{0}^{\infty }k\left(u\right)\left|h\left(z_{1}^{i}\left(s-u\right)\right)-h\left(z_{1}^{i-1}\left(s-u\right)\right)\right|du<1. \end{array}$$
It follows by (Equation 19) and (Equation 20) that
(21)$$\begin{array}{cc} |z_{2}\left(s\right)| & \leq \sum_{i=1}^{N}\left|z_{2}^{i}\left(s\right)-z_{2}^{i-1}\left(s\right)\right|<N \end{array}$$
and
(22)$$\begin{array}{cc} \; & {\left|a\left(s\right)-\left(b\left(s\right)-\xi \right)\xi \right|}_{0}|z_{1}\left(s\right)|+|f\left(z_{1}\left(s-\delta \left(s\right)\right)\right)|+\int_{0}^{\infty }k\left(u\right)\left|h\left(z_{1}\left(s-u\right)\right)\right|du\\ \; & \leq \sum_{i=1}^{N}{\left(|a\left(s\right)-\left(b\left(s\right)-\xi \right)\xi \right|}_{0}|z_{1}^{i}\left(s\right)-z_{1}^{i-1}\left(s\right)|+|f\left(z_{1}^{i}\left(s-\delta \left(s\right)\right)\right)-f\left(z_{1}^{i-1}\left(s-\delta \left(s\right)\right)\right)|\\ \; & +\int_{0}^{\infty }k\left(u\right)|h\left(z_{1}^{i}\left(s-u\right)\right)-h\left(z_{1}^{i-1}\left(s-u\right)\right)|du)+|f\left(0\right)|+h\left(0\right)\\ \; & {<N|a\left(s\right)-\left(b\left(s\right)-\xi \right)\xi |}_{0}+2N+\left|f\left(0\right)\right|+|h\left(0\right)|:=M. \end{array}$$
From (Equation 21) and (Equation 22), we have
(23)$$\begin{array}{cc} \left||\Phi z|\right| & \leq |\Phi_{1}{z|}_{0}+{\left|\Phi_{2}z\right|}_{0}\\ \; & \leq \hat{g}\int_{0}^{T}|z_{2}\left(s\right)|ds+\hat{\hbar }\int_{0}^{T}{\left(|a\left(s\right)-\left(b\left(s\right)-\xi \right)\xi \right|}_{0}\left|z_{1}\left(s\right)\right|\\ \; & +|f\left(z_{1}\left(s-\delta \left(s\right)\right)\right)|+\int_{0}^{\infty }k\left(u\right)\left|h\left(z_{1}\left(s-u\right)\right)\right|du)ds\\ \; & \leq \hat{g}TN+\hat{\hbar }MT. \end{array}$$
Furthermore, for t∈R, we have
(24)$$\begin{array}{cc} \frac{d\Phi z}{dt} & =(g\left(t,t+T\right)z_{2}\left(t+T\right)-g\left(t,t\right)z_{2}\left(t\right)-\xi \Phi_{1}z\left(t\right),\\ \; & h\left(t,t+T\right)F\left(t+T\right)-h\left(t,t\right)F\left(t\right)-\left(b\left(t\right)-\xi \right)\Phi_{2}z\left(t\right){)}^{T}\\ \; & =(-\xi \Phi_{1}z\left(t\right)+z_{2}\left(t\right),\\ \; & -\left[a\left(t\right)-\left(b\left(t\right)-\xi \right)\xi \right]z_{1}\left(t\right)+f\left(z_{1}\left(t-\delta \left(t\right)\right)\right)+\int_{0}^{\infty }k\left(u\right)\left|h\left(z_{1}\left(t-u\right)\right)\right|du-\left(b\left(t\right)-\xi \right)\Phi_{2}z\left(t\right){)}^{T}. \end{array}$$
By (Equation 23) and (Equation 24), we have
$$\begin{array}{cc} \left||\frac{d\Phi z}{dt}|\right| & \leq |\frac{d\Phi_{1}z}{dt}{|}_{0}+|\frac{d\Phi_{2}z}{dt}{|}_{0}\\ & \leq \xi |\Phi_{1}{z|}_{0}+|z_{2}{|}_{0}{+|a\left(t\right)-\left(b\left(t\right)-\xi \right)\xi |}_{0}{\left|z_{1}\right|}_{0}\\ & +|f\left(z_{1}\left(t-\delta \left(t\right)\right)\right){|}_{0}+|h\left(z_{1}\left(t-u\right)\right){|}_{0}{+|b\left(t\right)-\xi |}_{0}{\left|\Phi_{2}z\right|}_{0}\\ & \leq \xi \hat{g}TN+N+M+{\left|b\left(t\right)-\xi \right|}_{0}\hat{\hbar }MT. \end{array}$$
Hence {Φz:z∈K,||z||≤μ} is a uniformly bounded and equicontinuous function. Due to the Ascoli–Arzela theorem [20], the function Φ is completely continuous. □

**Theorem** **1.**
*Suppose that assumption (H${}_{1}$) holds. Furthermore, assume that there are positive constants r and R with r<R such that*

(25)
$$\underset{||\phi ||=r,\phi \in K}{\sup}\int_{0}^{T}\left|\phi_{2}\left(s\right)\right|ds\leq \frac{r}{2\hat{g}},$$


(26)
$$\begin{array}{cc} \underset{||\phi ||=r,\phi \in K}{\sup}\int_{0}^{T} & {\left(|a\left(s\right)-\left(b\left(s\right)-\xi \right)\xi \right|}_{0}|\phi_{1}\left(s\right)|+|f\left(\phi_{1}\left(s-\delta \left(s\right)\right)\right)|\\ \; & +\int_{0}^{\infty }k\left(u\right)\left|h\left(\phi_{1}\left(s-u\right)\right)\right|du)ds\leq \frac{r}{2\hat{\hbar }}, \end{array}$$


(27)
$$\underset{||\phi ||=R,\phi \in K}{\inf}\int_{0}^{T}\left|\phi_{2}\left(s\right)\right|ds\geq \frac{R}{2\overset{ˇ}{g}}$$

*and*

(28)
$$\begin{array}{cc} \underset{||\phi ||=R,\phi \in K}{\inf}\int_{0}^{T} & {\left(|a\left(s\right)-\left(b\left(s\right)-\xi \right)\xi \right|}_{0}|\phi_{1}\left(s\right)|+|f\left(\phi_{1}\left(s-\delta \left(s\right)\right)\right)|\\ \; & +\int_{0}^{\infty }k\left(u\right)\left|h\left(\phi_{1}\left(s-u\right)\right)\right|du)ds\geq \frac{R}{2\overset{ˇ}{\hbar }}. \end{array}$$

*Then, system (7) has a T-periodic solution z with r≤||z||≤R, i.e., Equation (Equation 1) has a T-periodic solution x>0.*


**Proof.** Let $z={\left(z_{1},z_{2}\right)}^{T}\in K$ and ||z||=r. From (Equation 25) and (Equation 26), we have
$$|\Phi_{1}{z|}_{0}\leq \hat{g}\int_{0}^{T}\left|z_{2}\left(s\right)\right|ds\leq \frac{r}{2}$$
and
$$|\Phi_{2}{z|}_{0}\leq \hat{\hbar }\int_{0}^{T}\left({\left|a\left(s\right)-\left(b\left(s\right)-\xi \right)\xi \right|}_{0}|z_{1}\left(s\right)|+|f\left(z_{1}\left(s-\delta \left(s\right)\right)\right)|+\int_{0}^{\infty }k\left(u\right)\left|h\left(z_{1}\left(s-u\right)\right)\right|du\right)ds\leq \frac{r}{2}.$$
Thus,
$$\left||\Phi z||\leq \right|\Phi_{1}{z|}_{0}+|\Phi_{2}{z|}_{0}\leq r=||z||.$$
where $z\in K\cap \partial \Omega_{1},\;\Omega_{1}=\left\{z\in X:||z||<r\right\}$. Similar to the above proof, in view of (Equation 26) and (Equation 27), we have
$$|\Phi_{1}{z|}_{0}\geq \overset{ˇ}{g}\int_{0}^{\omega }\left|z_{2}\left(s\right)\right|ds\geq \frac{R}{2}$$
and
$$|\Phi_{2}{z|}_{0}\geq \overset{ˇ}{\hbar }\int_{0}^{T}\left({\left|a\left(s\right)-\left(b\left(s\right)-\xi \right)\xi \right|}_{0}|z_{1}\left(s\right)|+|f\left(z_{1}\left(s-\delta \left(s\right)\right)\right)|+\int_{0}^{\infty }k\left(u\right)\left|h\left(z_{1}\left(s-u\right)\right)\right|du\right)ds\geq \frac{R}{2}.$$
Thus,
$$\left||\Phi z||\geq \right|\Phi_{1}{z|}_{0}+|\Phi_{2}{z|}_{0}\geq R=||z||,$$
where $z\in K\cap \partial \Omega_{2},\;\Omega_{2}=\left\{z\in X:||z||<R\right\}$. By Lemma 2, Φ has a fixed point *z* in $K\cap ({\bar{\Omega }}_{2}\setminus \Omega_{1})$. with r≤||z||≤R, hence, system (7) has a T-periodic solution *z* with r≤z≤R, i.e., Equation (Equation 1) has an *T*-periodic solution $z_{1}>0$. □

## 4. Positive Periodic Solution of Equation (Equation 2)

According to the proof of the existence of the positive periodic solution of Equation (Equation 1) and Lemma 1, we can easily obtain the existence of the positive periodic solution of Equation (Equation 2). Let
$$y\left(t\right)={\left(Ax\right)}^{\prime }\left(t\right)+\xi x\left(t\right),$$
where ξ>0 is a constant. Then, Equation (Equation 2) is changed into the following system:(29)$$\left\{\begin{array}{c} {\left(Ax\right)}^{\prime }\left(t\right)=-\xi Ax\left(t\right)-\xi c\left(t\right)x\left(t-\tau \right)+y\left(t\right),\\ y^{\prime }\left(t\right)=-\left(b\left(t\right)-\xi \right)y\left(t\right)-\left(b\left(t\right)-\xi \right)c^{\prime }\left(t\right)x\left(t-\tau \right)-\left(b\left(t\right)-\xi \right)c\left(t\right)x^{\prime }\left(t-\tau \right)\\ -\left[a\left(t\right)-\left(b\left(t\right)-\xi \right)\xi \right]x\left(t\right)+f\left(x\left(t-\delta \left(t\right)\right)\right)+\int_{0}^{\infty }k\left(s\right)h\left(x\left(t-s\right)\right)ds. \end{array}\right.$$
Since system (Equation 29) is equivalent to (Equation 2), we just have to study the existence of positive periodic solutions to system (Equation 29). Let
$$X=\{z\in C\left(R,R^{2}\right):z\left(t+T\right)=z\left(t\right),\;z={\left(Ax,y\right)}^{T}\}$$
with the norm $\left||z||=\max\{|Ax{|}_{0},|y{|}_{0}\right\}$. Then, X is a Banach space. Let
$$K=\{z\in X:\left(Ax\right)\left(t\right)\geq \theta |Ax{|}_{0}\;\mathrm{and}\;y\left(t\right)\geq \theta |y{|}_{0},\;t\in \left[0,T\right]\},$$
where $\theta =\min\{\frac{\overset{ˇ}{g}}{\hat{g}},\frac{\overset{ˇ}{\hbar }}{\hat{\hbar }}\}$, $\overset{ˇ}{g},\hat{g},\overset{ˇ}{\hbar },\hat{\hbar }$ are defined by (10) and (11). Integrate (Equation 29) from *t* to t+T and obtain that
(30)$$\left\{\begin{array}{c} (Ax\left(t\right)=\int_{t}^{t+T}g\left(t,s\right)\left[-\xi c\left(s\right)x\left(s-\tau \right)+y\left(s\right)\right]ds,\\ y\left(t\right)=\int_{t}^{t+T}\hbar \left(t,s\right)F\left(s\right)ds, \end{array}\right.$$
where
$$g\left(t,s\right)=\frac{e^{(s-t)\xi }}{e^{\xi T}-1},$$
$$\hbar \left(t,s\right)=\frac{\exp\left(\int_{s}^{t}\left(b\left(u\right)-\xi \right)du\right)}{\exp\left(\int_{0}^{T}\left(b\left(u\right)-\xi \right)du\right)-1},$$
$$\begin{array}{cc} F(t) & =-\left(b\left(t\right)-\xi \right)c^{\prime }\left(t\right)x\left(t-\tau \right)-\left(b\left(t\right)-\xi \right)c\left(t\right)x^{\prime }\left(t-\tau \right)\\ & -\left[a\left(t\right)-\left(b\left(t\right)-\xi \right)\xi \right]x\left(t\right)+f\left(x\left(t-\delta \left(t\right)\right)\right)+\int_{0}^{\infty }k\left(s\right)h\left(x\left(t-s\right)\right)ds. \end{array}$$
In view of (Equation 30), for each $z={\left(Ax,y\right)}^{T}\in X$, define an operator Φ:K→K as
$$\begin{array}{cc} \left(\Phi z)(t\right) & ={\left(\int_{t}^{t+T}g\left(t,s\right)\left[-\xi c\left(s\right)x\left(s-\tau \right)+y\left(s\right)\right]ds,\int_{t}^{t+T}\hbar \left(t,s\right)F\left(s\right)ds\right)}^{T}, \end{array}$$
where $\Phi z={\left(\Phi_{1}z,\Phi_{2}z\right)}^{T},\;g\left(t,s\right),\hbar \left(t,s\right)$, and F(s) are defined by (Equation 30). Thus, the existence of a positive periodic solution of system (Equation 29) is equivalent to finding the fixed point of operator Φ. Since the proofs of Lemmas 5 and 6 are similar to the proofs of Lemmas 3 and 4, we omit them.

**Lemma** **5.**
*The mapping Φ maps K into K.*


**Lemma** **6.**
*The mapping Φ:K→K is completely continuous.*


**Theorem** **2.**
*Suppose that assumption (H${}_{1}$) holds. Furthermore, assume that there are positive constants r and R with r<R such that*

$$\underset{||\phi ||=r,\phi \in K}{\sup}\int_{0}^{T}\left({\left|-\xi c\left(s\right)\right|}_{0}|A^{-1}\phi_{1}\left(s-\tau \right)|+|\phi_{2}\left(s\right)|\right)ds\leq \frac{r}{2\hat{g}},$$


$$\begin{array}{cc} \underset{||\phi ||=r,\phi \in K}{\sup}\int_{0}^{T} & {\left(|a\left(s\right)-\left(b\left(s\right)-\xi \right)\xi \right|}_{0}|A^{-1}\phi_{1}\left(s\right)|+|f\left(A^{-1}\phi_{1}\left(s-\delta \left(s\right)\right)\right)|\\ & +|\left(b\left(t\right)-\xi \right)c^{\prime }{\left(t\right)|}_{0}|A^{-1}\phi_{1}{\left(s-\tau \right)|+|\left(b\left(t\right)-\xi \right)c\left(t\right)|}_{0}\left|{\left(A^{-1}\phi_{1}\left(s-\tau \right)\right)}^{\prime }\right|\\ & +\int_{0}^{\infty }k\left(u\right)\left|h\left(A^{-1}\phi_{1}\left(s-u\right)\right)\right|du)ds\leq \frac{r}{2\hat{\hbar }}, \end{array}$$


$$\underset{||\phi ||=R,\phi \in K}{\inf}\int_{0}^{T}\left({\left|-\xi c\left(s\right)\right|}_{0}|A^{-1}\phi_{1}\left(s-\tau \right)|+|\phi_{2}\left(s\right)|\right)ds\geq \frac{R}{2\overset{ˇ}{g}},$$

*and*

$$\begin{array}{cc} \underset{||\phi ||=R,\phi \in K}{\inf}\int_{0}^{T} & {\left(|a\left(s\right)-\left(b\left(s\right)-\xi \right)\xi \right|}_{0}|A^{-1}\phi_{1}\left(s\right)|+|f\left(A^{-1}\phi_{1}\left(s-\delta \left(s\right)\right)\right)|\\ & +|\left(b\left(t\right)-\xi \right)c^{\prime }{\left(t\right)|}_{0}|A^{-1}\phi_{1}{\left(s-\tau \right)|+|\left(b\left(t\right)-\xi \right)c\left(t\right)|}_{0}\left|{\left(A^{-1}\phi_{1}\left(s-\tau \right)\right)}^{\prime }\right|\\ & +\int_{0}^{\infty }k\left(u\right)\left|h\left(A^{-1}\phi_{1}\left(s-u\right)\right)\right|du)ds\geq \frac{R}{2\overset{ˇ}{\hbar }}. \end{array}$$

*Then, system (Equation 29) has a T-periodic solution z with r≤||z||≤R, i.e., Equation (Equation 2) has a T-periodic solution x>0 provided that*

(31)
$$c\left(t\right)\geq 0\;\;\mathrm{for}\;c_{0}<1\;\mathrm{and}\;t\in R$$

*or*

(32)
c(t)<0forσ>1andt∈R.



**Proof.** Similar to the proof of Theorem 1, system (Equation 29) has a *T*-periodic solution $z={\left(z_{1},z_{2}\right)}^{T}$ such that $z_{1}=Ax>0$, i.e., $x=A^{-1}z_{1}.$ By Lemma 1, we have
(33)$$\left[A^{-1}z_{1}\right]\left(t\right)=\left\{\begin{array}{cc} z_{1}\left(t\right)+\sum_{j=1}^{\infty }\prod_{i=1}^{j}c\left(t-\left(i-1\right)\tau \right)z_{1}\left(t-j\tau \right),\;c_{0}<1,\;\forall z_{1}\in K, & \\ -\frac{z_{1}\left(t+\tau \right)}{c(t+\tau )}-\sum_{j=1}^{\infty }\prod_{i=1}^{j+1}\frac{1}{c(t+i\tau )}z_{1}\left(t+j\tau +\tau \right),\;\sigma >1,\;\forall z_{1}\in K. & \end{array}\right.$$
From (Equation 31)–(Equation 32), we have x>0 and Equation (Equation 2) has a positive *T*-periodic solution. □

**Remark** **1.**
*Green functions are crucial for studying the positive periodic solutions of second-order nonlinear Equations (Equation 1) and (Equation 2). However, there are no Green functions for (Equation 1) and (Equation 1) with periodic boundary condition and variable coefficients. This paper aims to propose a new method to study the above equations to avoid the difficulty of finding Green functions. We use the order reduction method to reduce the higher-order equation into a lower-order system, so we can avoid solving the Green function, and we can directly use the fixed point theorem to study the lower-order system.*


**Remark** **2.**
*In recent years, a huge amount of literature has come into existence for studying the positive periodic solution of neutral second-order nonlinear equations. Wu and Wang [14] studied the following second-order neutral equation:*

(34)
$${\left(x\left(t\right)-cx\left(t-\tau \right)\right)}^{\prime \prime }+a\left(t\right)x\left(t\right)=\phi b\left(t\right)f\left(x\left(t-\delta \left(t\right)\right)\right),$$

*where c∈(−1,0) is a constant and ϕ∈(0,1) is a constant. By the use of the fixed point theorem in cones, sufficient conditions for the existence of the positive periodic solution of (Equation 34) are established. When c∈(−1,1), Cheung et al. [15] discussed the existence of a positive periodic solution for the following second-order neutral differential equation:*

(35)
$${\left(x\left(t\right)-cx\left(t-\tau \left(t\right)\right)\right)}^{\prime \prime }+a\left(t\right)x\left(t\right)=f\left(t,x\left(t-\tau \left(t\right)\right)\right).$$

*Fore more results about (Equation 35), see, e.g., [16,17] and related references. In a very recent paper [18] using the Leray–Schauder fixed point theorem, Cheng, Lv, and Li studied the following equation:*

(36)
$${\left(Ax\left(t\right)\right)}^{\prime \prime }+b\left(t\right)x^{\prime }\left(t\right)+a\left(t\right)x\left(t\right)=f\left(x\left(t-\delta \left(t\right)\right)\right),$$

*where $c\left(t\right)\in C^{1}\left(R,R\right)$ is a T-periodic function. They obtained a range $c\left(t\right)\in (-\frac{a_{0}-b_{\infty }c_{\infty }^{\prime }}{a_{\infty }+\delta b_{\infty }+a_{0}},$$\frac{a_{0}-b_{\infty }c_{\infty }^{\prime }}{a_{\infty }+\delta b_{\infty }+a_{0}})$ for guaranteeing the existence of the positive periodic solution to (Equation 36). However, in the above papers, they used the properties of Green functions and neutral operators. In the present paper, we use the order reduction method to study second-order nonlinear differential equations with variable coefficients. We wish that the methods of the present paper can be used to study positive periodic solutions of neutral second-order nonlinear equations with variable coefficients.*


## 5. Examples

Consider the following equation of model (Equation 1):(37)$$x^{\prime \prime }\left(t\right)+b\left(t\right)x^{\prime }\left(t\right)+a\left(t\right)x\left(t\right)=f\left(x\left(t-\delta \left(t\right)\right)\right)+\int_{0}^{\infty }k\left(s\right)h\left(x\left(t-s\right)\right)ds,$$
where $b\left(t\right)=3.1\times 10^{-2}+\sin^{2}t,\;a\left(t\right)=1+\sin t,\;\delta \left(t\right)=0.1,\;k\left(s\right)=e^{-s},$
$$f\left(u\right)=\frac{0.1u}{1+u^{2}},\;h\left(u\right)=\frac{1}{1+u^{3}}\;for\;u\in R.$$
Let
$$y\left(t\right)=x^{\prime }\left(t\right)+\xi x\left(t\right),$$
where $\xi =2\times 10^{-2}>0$ is a constant. Then, Equation (Equation 29) is changed into the following system:(38)$$\left\{\begin{array}{c} x^{\prime }\left(t\right)=-2x\left(t\right)+y\left(t\right),\\ y^{\prime }\left(t\right)=-\left(b\left(t\right)-2\right)y\left(t\right)-\left[a\left(t\right)-\left(b\left(t\right)-2\right)2\right]x\left(t\right)+f\left(x\left(t-0.1\right)\right)+\int_{0}^{\infty }e^{-s}h\left(x\left(t-s\right)\right)ds. \end{array}\right.$$
From $b\left(t\right)-\xi =1.1\times 10^{-2}+\sin^{2}t>0$, assumption (H${}_{1}$) holds. By simple calculation, we have
$$\int_{0}^{\infty }k\left(s\right)ds=\int_{0}^{\infty }e^{-s}ds=1,\;T=2\pi ,$$
$$\overset{ˇ}{g}=\frac{e^{-T\xi }}{e^{\xi T}-1}\approx 6.667,\;\;\hat{g}=\frac{e^{T\xi }}{e^{\xi T}-1}\approx 8.518,$$
$$\overset{ˇ}{\hbar }=\frac{\exp\left({-T|b\left(t\right)-\xi |}_{0}\right)}{\exp\left(\int_{0}^{T}\left(b\left(u\right)-\xi \right)du\right)-1}\approx 5.9\times 10^{-3},\;\hat{\hbar }=\frac{\exp\left({T|b\left(t\right)-\xi |}_{0}\right)}{\exp\left(\int_{0}^{T}\left(b\left(u\right)-\xi \right)du\right)-1}\approx 168.45,$$
$$\theta =\min\left\{\frac{\overset{ˇ}{g}}{\hat{g}},\frac{\overset{ˇ}{\hbar }}{\hat{\hbar }}\right\}\approx 3.56\times 10^{-5}.$$
Choose r=1 and R=10. When $\phi ={\left(\sin t,0.01\cos t\right)}^{T}$, we have
$$\underset{||\phi ||=r,\phi \in K}{\sup}\int_{0}^{T}\left|\phi_{2}\left(s\right)\right|ds=0.04\leq \frac{r}{2\hat{g}}=0.0587,$$
$$\begin{array}{cc} \underset{||\phi ||=r,\phi \in K}{\sup}\int_{0}^{T} & {\left(|a\left(s\right)-\left(b\left(s\right)-\xi \right)\xi \right|}_{0}|\phi_{1}\left(s\right)|+|f\left(\phi_{1}\left(s-\delta \left(s\right)\right)\right)|\\ & +\int_{0}^{\infty }k\left(u\right)\left|h\left(\phi_{1}\left(s-u\right)\right)\right|du)ds=5.93\times 10^{-4}\leq \frac{r}{2\hat{\hbar }}=0.003. \end{array}$$
When $\phi ={\left(\sin t,10\cos t\right)}^{T}$, we have
$$\underset{||\phi ||=R,\phi \in K}{\inf}\int_{0}^{T}\left|\phi_{2}\left(s\right)\right|ds=40\geq \frac{R}{2\overset{ˇ}{g}}=0.75$$
and
$$\begin{array}{cc} \underset{||\phi ||=R,\phi \in K}{\inf}\int_{0}^{T} & {\left(|a\left(s\right)-\left(b\left(s\right)-\xi \right)\xi \right|}_{0}|\phi_{1}\left(s\right)|+|f\left(\phi_{1}\left(s-\delta \left(s\right)\right)\right)|\\ & +\int_{0}^{\infty }k\left(u\right)\left|h\left(\phi_{1}\left(s-u\right)\right)\right|du)ds=1576\geq \frac{R}{2\overset{ˇ}{\hbar }}=847. \end{array}$$
Thus, all assumptions of Theorem 1 hold. Hence, system (Equation 38) has a *T*-periodic positive solution, i.e., Equation (Equation 37) has a *T*-periodic positive solution *x*; the corresponding numerical simulation is presented in Figure 1.

## 6. Conclusions and Discussions

In the last past decades, nonlinear second-order differential equations with variable coefficients have found successful applications in scientific areas including quantum field theory, fluid mechanics, gas dynamics, and chemistry. Hence, there exists ongoing research interest in second-order differential equations with variable coefficients, including existence, stability, and oscillation, which have been obtained, see, e.g., [22,23,24,25]. In this paper, we develop a reducing order method for studying second-order differential equations with variable coefficients, avoiding the difficulty of finding Green functions.

The methods of this paper can be extended to investigate other types of second-order differential equations such as stochastic differential equations, impulsive differential equations, fractional differential equations, and so on. We hope some authors can use the methods provided in this article to conduct more in-depth research on various types of second-order differential equations with variable coefficients. 

## Figures and Tables

**Figure 1 entropy-24-01286-f001:**
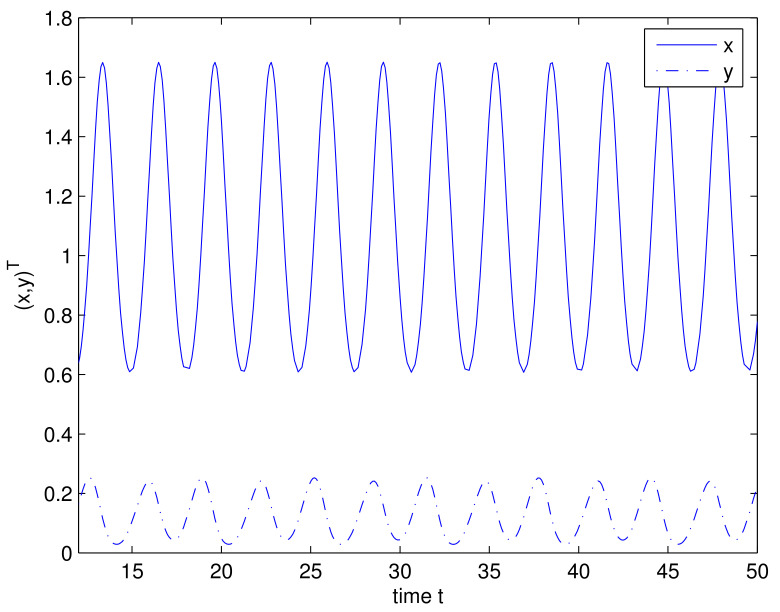
Periodic solution ${\left(x,y\right)}^{T}$ of system (Equation 38).

## Data Availability

Not applicable.

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
