# Peer review of "Positive Periodic Solution for Second-Order Nonlinear Differential Equations with Variable Coefficients and Mixed Delays"

_entropy, 2022, doi:10.3390/e24091286_

Round 1

Author Response

First of all, the authors would like to express their sincere
thanks to the Editor and the anonymous reviewers for their helpful
comments and suggestions. In this revision, all the comments from
the  reviewers have been seriously taken into account and
thoroughly implemented, and the explanation of the modifications
as well as corrections in this revision can be arranged as follows
(comment numbers are in 1:1 correspondence with the reviewers'
comments).

Reviewer 2 Report

1) What is the importance of the hypothesis |c(t)|\neq 0 in page 1?

2) The main background of this paper is not clear. Please kindly clarify the main background of this paper in the revision. The English should be improved throughout the whole paper and also the introduction part of the paper should be improved. This recent work could be mentioned : Convergence analysis on time scales for HOBAM neural networks in the Stepanov-like weighted pseudo almost automorphic space. Neural Computing and Applications, 1-15 (2020)

3) These results are there applicable for the environments investigated in the above paper (pseudo almost periodic, pseudo almost automorphic, etc...)?

4)  The simulations of the numerical example, should be added.

Author Response

(The authors gave the same response as above.)

Round 2

Reviewer 2 Report

Accept as it.

Author Response

We carefully revised the references  and added ORICD for Corresponding author。

See them in the revised paper.